# ATLAS: Actor-Critic Task-Completion with Look-ahead Action Simulation

## Abstract

We observe that current state-of-the-art web-agents are unable to effectively adapt to new environments without neural network fine-tuning, without which they produce inefficient execution plans due to a lack of awareness of the structure and dynamics of the new environment. To address this limitation, we introduce **ATLAS** (**A**ctor-Critic **T**ask-completion with **L**ook-ahead **A**ction **S**imulation), a memory-augmented agent that is able to make plans grounded in a *model* of the environment by simulating the consequences of those actions in *cognitive space*. Our agent starts by building a *"cognitive map"* by performing a lightweight curiosity driven exploration of the environment. The planner proposes candidate actions; the simulator predicts their consequences in cognitive space; a critic analyzes the options to select the best roll-out and update the original plan; and a browser executor performs the chosen action. On the **WebArena-Lite** Benchmark, we achieve a 63% success rate compared to 53.9% success rate for the previously published state-of-the-art. Unlike previous systems, our modular architecture requires no website-specific LLM fine-tuning. Ablations show sizable drops without the world-model, hierarchical planner, and look-ahead-based replanner confirming their complementary roles within the design of our system.

## 1 Introduction

Autonomous agents that can navigate and act on the web have the potential to perform complex tasks like information gathering, transactions, and site configuration on behalf of users (Yao et al., 2022; Sodhi, 2023). However, current web-based agents fall far short of human-level reliability on long-horizon tasks (Koh et al., 2024). The difficulty stems from partial observability, vast action spaces, the need for multi-step planning and memory in a web environment. For example, tasks in WebArena (Zhou et al., 2024) span diverse websites and require reasoning over multi-page navigation and content. An agent might be asked to "Tell me how many fulfilled orders I have over the past three days, and the total amount spent" on an e-commerce site, or "Set the homepage URL on my GitLab profile to https://egg.tart.com." Solving these tasks demands understanding site structure (e.g. admin dashboards, forms), remembering relevant information (like login states or filters applied), and avoiding irreversible mistakes (like deleting data or purchasing an item unintentionally).

Large language models (LLMs) have markedly improved semantic understanding and generation, suggesting they can enrich web navigation. Yet reliable long-horizon control remains elusive because LLM agents are typically reactive and lack structured memory and explicit planning. Current state-of-the-art web agents such as Plan-and-Act by Erdogan et al. (2025) have significant gaps - the agent's planner module is not grounded to the structure of the website and requires website-specific model fine-tuning of both planner and executor modules in order to enable new use-cases.

We address these gaps with ATLAS—an inference-time, actor–critic web agent that plans before acting via look-ahead simulation and retrieves structured memories to remain goal-directed over extended interactions. ATLAS has a modular architecture with four components: (1) a Planner that decomposes the task into subgoals; (2) an Actor that proposes diverse next-step candidates; (3) a Critic that forecasts each candidate's outcome by simulating state transitions and selects the safest, goal-advancing action; and (4) a multi-layer memory that is updated online and queried on demand. Together, these modules perform a simulated look-ahead tree search in conceptual space,

enabling adaptive environment-grounded planning, and efficient action selection in realistic, partially observable websites, including those with unexpected *environmental hazards*.

Our agent uses a hierarchical memory structure: (i) Working Memory for recent context; (ii) a Cognitive Map encoding state transitions and expected action outcomes; and (iii) Semantic Memory that captures environment-specific constraints (e.g. formats, hazards) in the form of world knowledge. The map is constructed via curiosity-driven exploration (MCTS-style trajectory mining) and agentic summarization that records action-to-outcome deltas in natural language, avoiding HTML bloat. During inference, the Actor conditions on the plan, retrieved memories, and in-context trajectories; the Critic filters out risky or myopic moves by simulating their consequences.

We summarize our contributions as the following:

- An *actor-critic planner* with LLM-based look-ahead that evaluates actions via simulated outcomes.
- A *multi-layer memory* with a cognitive map built through exploration and agentic summarization, used online for retrieval and replanning.
- A *practical modular architecture* that integrates *planning*, *memory*, and *simulation* to transform high-level instructions into safe, executable action sequences for long-horizon web tasks. Unlike previous systems, our system does not require website-specific LLM fine-tuning to ground to new websites and can thus easily be ported to new websites and new underlying LLMs.

## 2  RELATED WORK

**LLM-Based Autonomous Agents**  The integration of large language models into autonomous agents has revolutionized web navigation capabilities. Yao et al. (2023) pioneered the ReAct framework, demonstrating how LLMs can effectively combine reasoning and acting in interactive environments. This approach has been extended by Shinn et al. (2024) with Reflexion, which incorporates self-reflection mechanisms for improved decision-making over extended horizons.

**Web Navigation Agents**  Early web navigation systems relied predominantly on rule-based approaches and predefined scripts Liu et al. (2018), which, while interpretable, lacked the adaptability required for dynamic web environments. Recent advances have shifted toward learning-based methods, with several notable developments in autonomous web agents. Zhou et al. (2024) introduced WebArena, a comprehensive benchmark for evaluating web agents across realistic multi-step tasks, establishing a foundation for systematic evaluation in this domain. Building on this, Liu et al. (2024a) introduced WebArena-Lite, a curated subset addressing quality and scalability concerns in the original benchmark.

Recent work has focused on enhancing LLM agents with structured approaches to planning and memory and demonstrated significant progress: Qi et al. (2024) applied reinforcement learning principles to web navigation, achieving notable improvements through policy optimization. Zhang et al. (2025) developed WebPilot, focusing on multimodal understanding of web content, while Erdogan et al. (2025) introduced Plan-and-Act, emphasizing hierarchical task decomposition. Wang et al. (2024b) introduced Agent Workflow Memory (AWM), demonstrating the importance of persistent memory in multi-step web tasks. Yang et al. (2024) developed AgentOccam and demonstrated the effectiveness of simplifying the action space to natural language.

**Memory-Augmented Agents**  Memory mechanisms have emerged as crucial components for long-horizon task completion. Weston et al. (2014) established foundational work on memory networks, which has been adapted for sequential decision-making contexts. More recently, Zhong et al. (2023) proposed MemoryBank, a comprehensive framework for managing episodic and semantic memory in LLM-based agents. The concept of cognitive maps, originally from cognitive science Tolman (1948), has been adapted for artificial agents. Wayne et al. (2018) demonstrated neural implementations of cognitive mapping in reinforcement learning contexts, while Park et al. (2023) showed how LLMs can maintain and utilize spatial-temporal memory representations for complex behavioral simulation.

**Planning and Simulation in AI Agents**  Tree search and simulation-based planning have long been central to AI agent design. While classical approaches like Monte Carlo Tree Search Browne et al. (2012) have proven effective in discrete domains, recent work has extended these concepts to natural

language environments. Yao et al. (2024) introduced Tree of Thoughts, enabling LLMs to explore multiple reasoning paths through structured search.

World models, which enable agents to simulate future states without environment interaction, have gained renewed attention. Ha & Schmidhuber (2018) established foundational work on learned world models, while more recent efforts by Micheli et al. (2022) have shown how transformer architectures can serve as effective world models for sequential decision-making.

**Actor-Critic Methods and Look-ahead Planning**  Actor-critic architectures have proven particularly effective for combining policy learning with value estimation. While traditional actor-critic methods focus on reinforcement learning training Sutton & Barto (2018), recent work has adapted these principles to LLM-based agents. The integration of look-ahead planning with actor-critic frameworks has been explored in various contexts, with Silver et al. (2016) demonstrating the power of combining tree search with learned value functions in AlphaGo.

**Curiosity-Driven Exploration**  Exploration remains a fundamental challenge in autonomous agent design. Pathak et al. (2017) introduced intrinsic curiosity modules that drive exploration through prediction error, establishing a foundation for self-supervised exploration. Recent work has extended these concepts to language-based environments: Mu et al. (2024) demonstrated curiosity-driven exploration in embodied AI settings.

Our work builds upon these foundations by combining memory-augmented planning with look-ahead simulation in a modular architecture specifically designed for web navigation tasks. Unlike previous approaches that require environment-specific fine-tuning, ATLAS achieves strong performance through inference-time planning and memory retrieval, making it readily adaptable to new domains and underlying models.

## 3 METHOD

### 3.1 PROBLEM FORMULATION

We cast web navigation as a Partially Observable Markov Decision Process (POMDP) defined by the tuple $(\mathcal{S}, \mathcal{A}, \mathcal{O}, T, R)$, where $\mathcal{S}$ denotes the state set, $\mathcal{A}$ denotes the action set, $\mathcal{O}$ denotes the observation set, $T$ denotes the state transition function, and $R$ denotes the reward. Given a natural-language goal $q$, the agent must synthesize a plan and execute a sequence of actions $(a_0, \ldots, a_T)$ that reaches a goal-consistent terminal state. At each time step $t$, the agent receives partial observations $o_t \in \mathcal{O}$. Based on the observation $o_t$, the agent chooses an action to take $a_t \in \mathcal{A}$, such as `click` or `type`. The goal of the agent is to maximize the reward, i.e. fulfilling task $q$.

### 3.2 ARCHITECTURE OVERVIEW

ATLAS comprises four modules operating in an inference-time *actor–critic* loop with *action simulation in conceptual space*, shown in Figure 1 (a):

(1) **Planner** produces a high-level plan with subtasks, with the ability to replan;

(2) **Actor** proposes a small set of next-step candidates $C_t$;

(3) **Critic** performs outcome-aware simulation of each candidate and selects the safest, goal-advancing action;

(4) **Multi-layered memory** supplies working context, a cognitive map of state transitions, and world knowledge about the environment; it is queried online and updated as needed.

**Planner**  The planner analyzes and decomposes the natural language task $q$ into a structured plan with subtasks to finish. Given the initial observation $o_0$, the planner produces an initial plan $P_0$; at step $t$, it dynamically decides if the plan needs to be updated (replanning) given new evidence

$$P_0 = \text{Planner}(q, o_0), \qquad P_t = \text{Planner}(q, o_t, s_t, M). \tag{1}$$

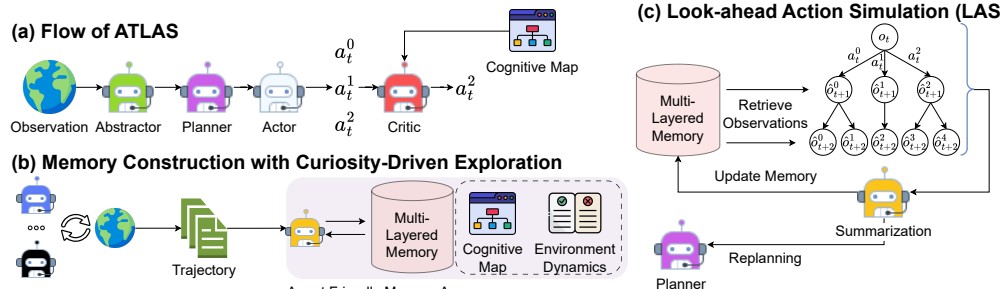

Figure 1: *Architecture of ATLAS.* (a) *Overall flow of ATLAS:* The raw observation $o_t$ is summarized to lower cognitive load. Then the planner makes a plan $P_t$ based on the summarized observation $o'_t$. The actor proposes $N$ possible candidate actions for next step. The critic provides judgment of action candidates and finalizes the best action $a_t$ to take by considering action outcomes obtained from the cognitive map. (b) *Memory construction with curiosity-driven exploration:* We build cognitive map by employing exploratory lightweight agents to interact with the environment. (c) *Look-ahead Action Simulation (LAS):* At each step, *ATLAS* simulates all candidate actions with the observation from the cognitive map, providing ability to look-ahead. We employ the memory agent to learn from LAS trajectories to make a better plan and update memory if necessary.

Plans are concise lists of sub-goals with success predicates (e.g., "Reports → Sales → Set dates → Read table"). Planner outputs are included in the context for the actor and critic. Our planner is implemented in the style of (Chae et al., 2025) and extended and described in 3.5.

**Actor-Critic interplay with look-ahead**  In our framework, at each step $t$, the actor proposes $N$ executable candidates with reasoning, and the critic evaluates it based on a value function $V(a)$.

$$C_t = \text{Actor}(q, P_t, o_t, s_t, M), \qquad |C_t| = N. \tag{2}$$

The critic evaluates each candidate action $a_t^i \in C_t$ and selects the best next action

$$a_t = \arg\max_{a \in C_t} V(a \mid q, P_t, o_t, s_t, M). \tag{3}$$

The utility estimate $V(a)$ is derived via LLM-based assessment that incorporates goal alignment, state viability (recoverability), action coherence, plan consistency, and outcome risk (e.g., destructive or dead-end transitions). Unlike previous efforts that attempt to learn implicit world-model of the environment by fine-tuning a neural network model, such as Chae (2024), we leverage the cognitive map to retrieve action outcomes of each action candidate (Section 3.3). This gives the agent system the ability to look ahead of the current step. We later extend this standard behavior with simulated tree search to further enhance exploratory ability in Section 3.4.

**Multi-layered memory**  We use three complementary memories:

- **Working Memory**: A task-specific memory wherein facts and observations are optionally stored into the LLM context for use during a particular episode.
- **Cognitive Map**: A graph of transitions $M = \{(o, a, o')\}$ with *agentic summaries* that store deltas and new affordances (e.g., "click `Reports` reveals {Sales, Products,... }") rather than raw HTML. The map supports retrieval $\hat{o}_{t+1} = M(o_t, a)$ for simulation and planning (Section 3.3).
- **Semantic Memory (World Knowledge)**: Learned environment dynamics (e.g., date/format rules, non-recoverable states), used to penalize risky actions and inform simulation (Section 3.3).

### 3.3 Memory construction via Curiosity-Driven Exploration

**Motivation**   Existing agents may fail because they are 1) not aware of the potential action outcomes, such as placing an order on a shopping website likely to lead to difficulty of canceling and refunding, and, 2) not familiar with environment-specific requirements, such as date format and search format (e.g. AgentOccam-Judge on WebArena-Lite Task 0) (Yang et al., 2024). This is a major gap between LLM agents and human intelligence, since humans can easily predict the outcome of an action with our world knowledge. Encouraging agents to explore the environment and store the findings in memory can effectively avoid actions leading to undesirable outcomes.

**Memory construction**   Prior work on artificial curiosity Pathak et al. (2017) has demonstrated the ability of neural networks operating in an agentic context to frame curiosity as a self-supervised learning task. Inspired by this insight, we augmented our agent with an artificial curiosity module to initialize the agent. Before evaluation, we perform a curiosity-driven exploration of the web environment to seed the cognitive map and world knowledge of the agent.

The memory construction process consists of the following steps:

- **Exploration policies**: We first launch a series of lightweight explorer subagents with diverse LLM generation temperatures and exploration policies. We embed coverage incentives into the prompt of the explorer agents. No task-completion reward is used, in order to avoid information leakage from the test set. We balance breadth, depth, and entropy, limiting the explorers to visit the most promising states within a fixed memory budget.

- **LLM based trajectory-mining**: Given the exploration trajectories, we employ an LLM to convert the trajectories to agentic summaries of the environmental transitions, and store it as a cognitive map. Additionally, we employ an LLM to produce agentic summaries of site-specific rules, constraints, and hazards in semantic memory.

**Memory layer 1: Cognitive Map**   The cognitive map encodes structured knowledge about the environment's dynamics, including state transitions and causal relationships. Conceptually, it is akin to a learned world model or transition model in reinforcement learning, capturing how actions alter observations. For example, "clicking `Add to Cart` on a product page" results in a cart update notification, while "entering text in the search bar" leads to a results page. Formally, the cognitive map is represented by tuples $(o_t, a_t, o_{t+1})$, where $o_t$ and $o_{t+1}$ denote observations (e.g., HTML content or URLs in text-based web environments) at steps $t$ and $t+1$, and $a_t$ is the action executed at step $t$.

At each step of exploration, we document the current observation $o_t$, the executed action $a_t$, and the subsequent observation $o_{t+1}$. To enhance interpretability and lower cognitive load of the agent, we adopt an *agentic memory* strategy, where an LLM agent curates what is written into memory. Specifically, the memory agent produces concise summaries emphasizing (in addition to the raw observations):

- Differences between successive observations $(o_t, o_{t+1})$;
- Newly available actions in $o_{t+1}$ after executing $a_t$.

For retrieval, the cognitive map is queried with $(o_t, a_t)$, returning the next raw observation $o_{t+1}$, as well as LLM summaries. This design balances fidelity (retaining raw states) with abstraction (summarized transitions), enabling efficient reasoning over complex, text-based environments. When the retrieval hits an unexplored node in the cognitive map, a generic-placeholder observation is returned.

**Memory layer 2: Semantic Memory (World Knowledge)**   This memory captures environment-specific knowledge, such as constraints, formats, and idiosyncratic behaviors unique to each website. For instance, it encodes facts like "the date picker only accepts input in `MM/DD/YYYY` format" or "the admin portal does not support exporting tables into CSV files". By recording these particulars from prior explorations, semantic memory serves as a bridge between specific past experiences and working memory, which maintains awareness of the immediate environmental context. This integration enables agents to adapt more effectively to recurring interface patterns and site-specific

limitations. The cognitive map and semantic memory are also optionally updated online when execution encounters unseen transitions or world dynamics.

## 3.4 LOOK-AHEAD ACTION SIMULATION (LAS)

The standard actor-critic interplay (Section 3.2) is a good baseline but may suffer from insufficient exploration and lack of foresight. To mitigate this issue, we propose Look-ahead Action Simulation (LAS).

At step $t$, the actor first generates the action candidates $C_t$ as described in Equation 2. For each action candidate $a_t^i \in C_t$, the critic hypothetically selects $a_t^i$ as the final action to execute and provides criticism. The resulting change of observation is retrieved from the cognitive map

$$\hat{o}_{t+1}^i = M(o_t, a_t^i), \tag{4}$$

where $i$ is the index of the action candidate at step $t$.

We repeat this for $D$ times, resulting in a set of rolled-out trajectories of length $D$. Let $\hat{\tau}$ denote a simulated trajectory (length $D$) with value $V(\hat{\tau})$. We apply confidence weighting based on transition uncertainty $U(s, a)$:

$$\tilde{V}(\hat{\tau}) = V(\hat{\tau}) \cdot \prod_{(s,a) \in \hat{\tau}} \big(1 - U(s, a)\big). \tag{5}$$

The best trajectory determines the real action $a_t$.

**Comparison to prior work** Existing agents perform tree search with LLMs as a reward function or a world model and measures the quality of each possible action candidate with a numerical score. Then the agent only executes actions whose scores are above a certain threshold. Our simulated tree search has three advantages compared to prior methods:

1. *Trustworthiness*: Prior works rely on LLM to envision outcomes of actions. Since LLMs are not explicitly trained to be a world model, this behavior is prone to hallucination and is not robust enough. In contrast, our method leverages real observations that are much more trustworthy.

2. *Comprehensiveness*: Prior works essentially conducts a greedy search (one-step), with low-scoring branches directly pruned without further consideration. Some actions may not be good at the immediate step $t$ but useful at the next step $t + 1$. Such actions may be overlooked in existing agent systems. While our method is similar to beam search (multi-step), considering the joint outcome of a sequence of actions.

3. *Efficiency*: Our exploration is a simulation in conceptual space, which is much more efficient than actually executing the actions. It also avoids stateful actions that cannot be recovered, since no action is executed.

## 3.5 LOOK-AHEAD ACTION SIMULATION-BACKED DYNAMIC REPLANNING AND MEMORY UPDATE

**Replanning** We dynamically trigger replanning when observations diverge from expectations:

$$\text{replan} = \mathbb{1}\big[\, \|o_t^{\text{obs}} - \hat{o}_t^{\text{exp}}\| > \varepsilon \,\big]. \tag{6}$$

A task-relevant plan requires a high-level view of the environment and the ability to foresee what would happen in future steps. We attempt to distill the foresight enabled by simulated tree search (Section 3.4), by using the result of the search to update our planner.

As Figure 1 (c) illustrates, the planner integrates a brief *exploration digest* (what worked/failed, newly exposed affordances, uncovered prerequisites) distilled by the memory writer, then updates $P_t$. This can be viewed as a highly simplified implementation of a basic causal learning module that attempts to update our causal model of the world - a highly simplified version of the conceptual flow introduced by Sontakke et al. (2021). This mechanism also prevents catastrophic forgetting of important context, which can happen if the replanner is run with every execution step.

Table 1: Evaluation Results for ATLAS versus other methods reported on WebArena-Lite. Best performance is in **bold**

| Agent | Avg w/ Multi-site | Avg w/o Multi-site | Gitlab | Reddit | Shopping | Shopping Admin | Maps | Multi-Site |
|---|---|---|---|---|---|---|---|---|
| WebPilot + GPT-4o | - | 35.3 | 39.4 | 65.1 | 36.9 | 24.7 | 33.9 | – |
| AWM + GPT-4-0613 | - | 33.0 | 31.8 | 50.9 | 30.8 | 29.1 | 43.3 | – |
| WebRL | - | 48.1 | 50.0 | 78.9 | 44.4 | 54.3 | 40.0 | – |
| Plan-and-Act | 53.9 | 57.5 | 53.3 | 84.2 | **55.6** | 48.6 | 46.6 | 30.0 |
| AgentOccam *(Claude-4-Sonnet)* | 47.9 | 51.0 | 66.7 | 63.2 | 40.0 | 54.3 | 23.1 | 40.0 |
| **ATLAS (Ours)** | **63.0** | **67.1** | **73.3** | **84.2** | 53.3 | **77.1** | **42.3** | **40.0** |

**Memory Update**    In addition to replanning, the agent must also be capable of updating its memory during action simulation. This process applies to both the cognitive map and episodic memory, ensuring that newly encountered patterns, constraints, or dynamics are incorporated into long-term knowledge. Crucially, decisions about what to retain, update, or forget are delegated to the memory agent, which curates information based on task relevance and environmental novelty. Such selective updating is particularly important during curiosity-driven exploration, where novel experiences can refine the agent's representation of the environment while preventing memory overload with redundant or irrelevant details.

## 4    EXPERIMENTAL SETUP

WebArena (Zhou et al., 2024) presents a realistic simulation environment comprising a broad array of web navigation tasks such as content retrieval, task execution, and form completion. Tasks vary in complexity, ensuring that the agent's capabilities are thoroughly tested in realistic scenarios - ranging from purchasing items for eCommerce shopping to updating GitLab code repositories.

The original WebArena consists of 811 tasks, however many of these cannot be performed - humans could only perform 78% of the WebArena. WebArena-Lite is a quality-controlled smaller subset of WebArena consisting of 165 tasks introduced by (Liu et al., 2024b) and has been adapted by prior work in the web agent space such as WebRL (Qi et al., 2024) and Plan-and-Act (Erdogan et al., 2025) as a higher quality and more scalable benchmark for evaluating web agents in the most realistic setting possible - incorporating realistic scenarios such as unexpected environment failures.

## 5    RESULTS

### 5.1    COMPARISON WITH OTHER BASELINES

In this section, we compare our model to other published results on the WebArena-Lite dataset. These are WebPilot (Zhang et al., 2025), Agent Workflow Memory (AWM) (Wang et al., 2024a), WebRL Qi et al. (2024), and Plan-and-Act (Erdogan et al., 2025). Our own work builds on top of AgentOccam (Yang et al., 2024) - we rerun our results on AgentOccam using the state-of-the-art LLM available to us for experimentation at scale, Claude-4 Sonnet.

### 5.2    ABLATION STUDY

In this section we conduct an ablation study where we study the effects of the different components of the system. Starting with AgentOccam as our base agent, we demonstrate that augmenting the agent with the direct HTML cognitive map *(Base + CM-Raw)* initially reduced performance but led to a dramatic improvement in performance after we enabled agentic summarization (Base + CM).

Table 2: Ablation Study Results for Individual Components of **ATLAS**.

| Agent | Avg w/ Multi-site | Avg w/o Multi-site | Gitlab | Reddit | Shopping | Shopping Admin | Maps | Multi-site |
|---|---|---|---|---|---|---|---|---|
| Plan-and-Act | 53.9 | 57.5 | 53.3 | 84.2 | **55.6** | 48.6 | 46.6 | 30 |
| AgentOccam *(Base)* | 47.9 | 46.7 | 66.7 | 68.4 | 40 | 42.9 | 30.8 | 30 |
| **Cognitive Map** | | | | | | | | |
| *Base + CM-Raw* | 44.8 | 47.1 | 70 | 68.4 | 35.6 | 51.4 | 19.2 | 0 |
| *Base + CM* | 57.4 | 55.8 | 76.7 | 78.9 | 46.7 | 71.4 | 19.2 | 30 |
| **Planning** | | | | | | | | |
| *Base + HL* | 50.9 | 54.2 | 63.3 | 78.9 | 53.3 | 57.1 | 15.4 | 20 |
| **ATLAS** *Base+CM+HL+LA* | **63.0** | **67.1** | **73.3** | **84.2** | 53.3 | **77.1** | **42.3** | **40.0** |

In addition, integrating a high-level planner *(Base + HL)* in the style of Chae et al. (2025) also improves performance on top of the base agent.

Finally, we construct the final ATLAS agent by integrating both the cognitive map and high-level planner and further extend the system to include replanning via look-ahead search *(Base + CM + HL + LA)* in order to condition the planner on the cognitive map, we see that the two systems demonstrate complementary performance to produce superior performance in conjunction - achieving state of the art results on WebArena-Lite.

## 6  CONCLUSION

This work presented **ATLAS**, a web navigation agent that couples explicit, structured memory with hierarchical planning to turn open-ended browsing into a sequence of verifiable, low-entropy decisions. By leveraging contemporary large language models within a modular control loop, the agent maintains situational awareness across pages, decomposes goals into intermediate subgoals, and adapts its strategies as the interface or task constraints evolve. The result is a system that is not only more sample-efficient and time-efficient during exploration, but also more interpretable: intermediate memories, subplans, and decision rationales expose where and why the agent changes course.

**Future Work**   Looking forward, we hope to see a research agenda that emphasizes *principled* generalization of this work rather than tuning performance on a single-benchmark.

* First, our world-model representation of the web is still in its infancy. We hope to see others develop web-native world models that abstract repeated patterns (e.g., filters, tables, forms) into sub-programs and support counterfactual "what-if" reasoning, not merely retrieval.

* Second, next-generation planning should be budget-aware and safety-aware by design, trading off success, latency, and risk through calibrated uncertainty and constraint handling.

* Third, system robustness needs to be measured—not assumed—via stress tests that include UI drift, authentication flows, stochastic failures, and long-horizon, multi-session tasks.

* Finally, as agentic systems start to close the gap with human performance, evaluation looking forward must move beyond pass/fail to incorporate cost of computation, side-effect penalties, reproducibility across seeds, and transparency of the intermediate state.

Taken together, these directions aim at agents that learn enduring abstractions of the web, plan under explicit budgets and constraints, and expose interpretable interfaces for verification and collaboration. We view this separation of concerns—memory, planning, and control—as a durable scaffold for the next generation of reliable, adaptable web agents which are certain to become ubiquitous and invaluable tools in the years to come.

ETHICS STATEMENT

This work focuses on improving the task execution ability of LLM agents, which can potentially have a larger impact on democratizing large models and facilitate routine tasks. All experiments are conducted on publicly available datasets following the code of ethics.

REPRODUCIBILITY STATEMENT

Our experiments are based on public implementations and API-based LLMs. We will release our code to the community.

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

# A AGENT PROMPTS

## A.1 PLANNER PROMPT

You are an AI assistant tasked with generating structured checklists that highlight
subgoals necessary to complete a task.
## Task Description
User Instruction (Goal): {task_objective}
Start Website URL: {initial_url}
Initial observation: {initial_html}
Guidelines for Checklist Generation
1. Identify Essential High−Level Subgoals:
− A subgoal should represent a significant step involving user interaction that leads
noticeable page transitions or meaningful changes in system state.
− Consolidate closely related user actions (such as applying multiple filters or sele
− Prioritize only the most critical interactions necessary for meaningful progression
avoiding the inclusion of minor or unnecessary steps (e.g., scroll, hover).
2. Provide a Concise Subgoal Analysis:
− Before creating the checklist, offer a brief paragraph summarizing the main subgoal
emphasizing significant transitions or page−level interactions.
3. Ensure Clear Goal:
− If multiple related interactions occur (e.g., setting filters 1, 2, and 3), combine
subgoal with clear criteria verifying all required conditions.
− The checklist should contain only essential steps, explicitly excluding unnecessary
actions, and should not exceed five critical subgoals. It is not necessary to use al
checklist items if fewer steps adequately represent the essential subgoals.
### Output Format
Before generating the checklist, first produce a concise subgoal analysis in a single
paragraph summarizing the required interactions. Then, based on this, generate the c
following the format below:

## A.2 REPLANNING PROMPT

## A.3 ACTOR PROMPT

You are an AI assistant performing tasks on a web browser. You will be provided with

Generate the response in the following format:
{output_specifications}

You are ONLY allowed to use the following action commands. Strictly adheres to the g
If you think you should refine the plan, use the following actions:
branch [parent_plan_id] [new_subplan_intent]: To create a new subplan based on PREVIO
prune [resume_plan_id] [reason]: To return to a previous plan state when the current

Otherwise, use the following actions:
click [id]: To click on an element with its numerical ID on the webpage. E.g., 'click
go_back: To return to the previously viewed page.
go_home: To return to the homepage where you can find other websites.
note [content]: To take note of all important info w.r.t. completing the task to ena
stop [answer]: To stop interaction and return response. Present your answer within th
type [id] [content] [press_enter_after=0|1]: To type content into a field with a spe

## A.4 CRITIC PROMPT

594 You are a seasoned web navigator. You now assess the value and risk of serveral web

595

596 Adhere to the following output format:

597 {output_specifications}

598

599 Note that 'branch' and 'prune' are planning actions that will modify the PREVIOUS PLA

600 "input_template": '''The following is the interaction history, current state, and ac

601

602 A.5 COGNITIVE MAP PROMPT

603

604

605

606 A.6 EPISODIC MEMORY PROMPT

607

608 You are an expert in summarizing agent exploration.

609 You are given a list of exploration trajectories and a previous environment dynamics

610 Your task is to update the environment dynamics summary by incorporating the new evid

611 1. what is supported in the environment

612 2. what is prohibited / invalid in the environment

613 Now update the environment dynamics summary by incorporating the new evidence.

614 Do not discard prior information unless it is contradicted. Expand, refine, or adjus

615

616 ## Output must include:

617

618 1. Allowed Actions

619 - Actions that successfully changed the state or were permitted by the environment.

620 - Note under what conditions they became available (e.g., ''checkout button only afte

621

622 2. Prohibited / Invalid Actions

623 - Actions attempted by the agent that failed (blocked, ignored, caused an error, or l

624 - Capture patterns of prohibition (e.g., ''cannot add item before selecting size'').

625

626 3. Environment specific formats

627 - What is the date format

628 - What is the search format

629 - What is the URL format

630

631 4. Newly Exposed Options

632 - Actions that became visible or available during exploration (e.g., new buttons, me

633 - Indicate whether these seem critical or peripheral.

634

635 5. Environment Reliability

636 - Cases where the environment behaved inconsistently or failed to respond properly.

637 - Errors, misleading cues, or missing elements.

638

639 6. Coverage & Unknowns

640 - What rules/dynamics are now confirmed.

641 - What remains uncertain or untested.

642

643 ## Output Format

644 - Updated Environment Dynamics

645 - Allowed Actions

646 - Prohibited / Invalid Actions

647 - State Transition Dynamics

- Newly Exposed Options

- Environment Reliability

- Coverage & Unknowns

## Notes
Keep the summary concise but precise (less than 5 bullets per section).
Use no more than 500 tokens.
If new findings contradict older ones, mark the correction explicitly. Otherwise, do
Only describe environment dynamics.

## Exploration Trajectories
{exploration_trajectories}

## Previous Summary
{prev_summary}

