# OpenReview forum: "ATLAS: Actor-Critic Task-Completion with Look-ahead Action Simulation"
_ICLR.cc/2026/Conference — ICLR 2026 Conference Desk Rejected Submission_

### Official Review · Reviewer_HeKd · 2025-10-23

**Soundness:** 2
**Presentation:** 2
**Contribution:** 2
**Rating:** 2
**Confidence:** 4

**Summary:**

This paper proposes a new web agent framework based on LLMs to improve browser-use performance. Specifically, the authors proposes ATLAS, a new framework by prompting LLMs to 1) create a "cognitive map" of the corresponding tasks by using the LLM to explore the target websites; 2) use the precomputed cognitive map to enhance planning by employing a best-of-N strategy (prompting an LLM to propose some actions/trajectories, which are then evaluated by prompting the LLM again). The authors evaluated their method on WebArena-Lite and showed that it outperformed previous best method.

**Strengths:**

- I agree that current web agents lack web/environment understanding, and constructing a knowledge graph via exploration is an interesting method to help address the problem.
- The proposed framework (ATLAS) achieves performance improvement on WebArena-Lite compared to previous best model.
- The authors conducted careful ablation studies to justify the performance contribution from each proposed module in ATLAS.

**Weaknesses:**

1. ATLAS is a method that performs look-ahead planning (using cognitive map to perform simulated rollout, judged by an LLM to select the best action before action). This is highly similar to search algorithms such as MCTS (which is mentioned in the paper) but in the experiments no comparisons were made between ATLAS and any search-based methods. Many relevant baselines are thus missing from this paper, including [1-4].

2. ATLAS incurs substantial costs compared to other baselines in Table 1, yet the authors did not report the additional cost/compare against simple methods of similar cost to establish a fair comparison. Specifically, since ATLAS requires 1) calling LLMs many times before testing to construct cognitive map; 2) calling LLM-based planner, actor, and critic multiple times to create multiple trajectories, from which the best is used for execution. Thus, I believe at least the authors should compare against a best-of-N baseline, where the N is chosen to achieve equal-compute compared to ATLAS.

3. The authors introduced many components in ATLAS based on heuristics (e.g., memory with three different layers, action selection via "confidence weighted uncertainty" of prompt-based value function, etc) but only evaluated ATLAS on WebArena-Lite. It is thus questionable whether/which proposed component is really critical and generalizable for other web benchmarks such as Mind2Web2.

4. The claim that ATLAS “simulates the consequences of those actions in cognitive space” (L17, Figure 1 caption, L305–307) appears overstated given the described method. As stated in Section 3.3 and L296–300, the cognitive map is constructed through (lots of) *real environment interactions/executions*, and Equation (4) indicates that “simulation” is effectively implemented by retrieving a cached next state from the real executions for a given state–action pair. This raises doubts about whether this process constitutes genuine simulation, as it depends heavily on prior real-world executions. In this sense, ATLAS’s look-ahead planning resembles conventional search-based approaches (e.g., [3], [4]), which are missed baselines in the experiment section.

---

References:

[1] Yao, Shunyu et al. “Tree of Thoughts: Deliberate Problem Solving with Large Language Models.” ArXiv abs/2305.10601 (2023): n. pag.

[2] Zhou, Andy et al. “Language Agent Tree Search Unifies Reasoning Acting and Planning in Language Models.” ArXiv abs/2310.04406 (2023): n. pag.

[3] Koh, Jing Yu et al. “Tree Search for Language Model Agents.” Trans. Mach. Learn. Res. 2025 (2024): n. pag.

[4] Yu, Xiao et al. “ExACT: Teaching AI Agents to Explore with Reflective-MCTS and Exploratory Learning.” ArXiv abs/2410.02052 (2024): n. pag.

**Questions:**

- in Equation 1 why is there $s_t$ as well as $o_t$? I believe $s_t$ should not be available for web benchmarks?
- in Equation 1, why is there no $P_t$ not a function of $P_{t-1}$, assuming that replanning is based on previous plans?
- while world modeling is not a critical aspect of this work, I believe the authors may have overlooked many existing work on LLM as world models for web agents. Examples include [1-2]. I believe these work should have directly addressed the limitations mentioned in L417-419.
- Equation 5 mentions a new quantity called "transition uncertainty U(s,a)", but it is never explained in the paper how that quantity is calculated.
- Paper's appendix is formatted incorrectly. Many texts are flowing over the page.


---

References:

[1] Gu, Yu et al. “Is Your LLM Secretly a World Model of the Internet? Model-Based Planning for Web Agents.” ArXiv abs/2411.06559 (2024): n. pag.

[2] Fang, Tianqing et al. “WebEvolver: Enhancing Web Agent Self-Improvement with Coevolving World Model.” ArXiv abs/2504.21024 (2025): n. pag.

---

### Official Review · Reviewer_24nw · 2025-10-30

**Soundness:** 2
**Presentation:** 2
**Contribution:** 2
**Rating:** 4
**Confidence:** 3

**Summary:**

The paper introduces ATLAS (Actor-Critic Task-completion with Look-ahead Action Simulation), a memory-augmented web agent that can adapt to new environments without fine-tuning. ATLAS builds a cognitive map of its environment through lightweight, curiosity-driven exploration and plans actions by simulating their outcomes in cognitive space. Its architecture combines a planner, a simulator, a critic, and an executor to refine action sequences iteratively. On the WebArena-Lite benchmark, ATLAS outperforms the previous state-of-the-art methods, while requiring no website-specific fine-tuning. Ablation studies demonstrate that components such as the world model, hierarchical planner, and look-ahead replanner are all crucial to the system’s performance.

**Strengths:**

1. The manuscript is well written, with figures that clearly illustrate the proposed methodology and equations that effectively present its technical foundations.

2. The experimental setup is carefully designed, featuring appropriate choices of baselines. The ablation study offers a comprehensive analysis of the contribution of each component to the overall performance.

**Weaknesses:**

1. Although ATLAS appears to be specifically tailored for the WebArena environment, the paper would benefit from evaluating the agent in more diverse and generalized scenarios. Demonstrating its effectiveness across a broader range of interactive environments would strengthen the claim of generality and provide deeper insights into the agent’s adaptability and robustness beyond the WebArena benchmark.

2. The experimental settings are not described with sufficient clarity. For instance, the number of random seeds used to compute the averaged results and standard deviations is not specified, which raises concerns about the statistical significance and robustness of the reported findings. Furthermore, the paper does not provide an adequate discussion of the time efficiency or computational cost of the proposed method—an important consideration for the practical deployment of modern intelligent agents. Including these details would enhance the reproducibility and credibility of the experimental evaluation.

**Questions:**

1. Can the proposed agent’s generalizability be evaluated on a broader range of environments?

2. Could the authors present the results with greater statistical rigor, specifying details such as the number of random seeds used and the corresponding standard deviations?

3. Would it be possible to include a time analysis—such as total training and inference time—and compare it against other baseline agents to assess computational efficiency?

---

### Official Review · Reviewer_wke9 · 2025-10-31

**Soundness:** 2
**Presentation:** 1
**Contribution:** 2
**Rating:** 2
**Confidence:** 4

**Summary:**

The paper presents ATLAS, an actor–critic web agent that performs look-ahead action simulation using a multi-layer memory system to improve planning and adaptability in web environments.
By combining a cognitive map, semantic memory, and hierarchical planner, ATLAS can simulate future action outcomes and dynamically replan without fine-tuning.

**Strengths:**

1. The authors propose ATLAS, an actor–critic framework with look-ahead action simulation for web agents, which achieves strong performance on the WebArena benchmark and demonstrates improved planning and reliability compared to prior methods.

**Weaknesses:**

1. The experimental scope is limited. Although the authors claim that ATLAS is generalizable across various websites, the evaluation is restricted to WebArena. Testing on more diverse and realistic web benchmarks such as WebVoyager [1] would strengthen the claim of generality.

2. The method section lacks clarity and organization. Definitions often appear long after their first use (e.g., $M$ in Section 3), and the section mixes motivation, related work, and algorithms without a clear logical flow. A more structured rewrite is needed.

3. The paper’s novelty appears incremental. The proposed framework seems to be an additive combination of two relatively standard components, planner and memory, without clear evidence of deeper integration or novel mechanisms.

4. The details of the method are insufficient. Key components such as the memory update mechanism (Section 3.5) are only briefly mentioned without details, and the interaction between memory and planning is not clearly described. It is also unclear how the “look-ahead” actions beyond step $t$ are generated.

5. (minor) Some formatting and writing issues remain, including inconsistent variable definitions and minor citation formatting errors (e.g., l226).

[1] WebVoyager: Building an End-to-End Web Agent with Large Multimodal Models.

**Questions:**

1. How does the “look-ahead” process obtain future actions beyond step $t$? I can only understand the the action at step $t$ is generated by the actor.

2. The memory module is described as not directly tied to planning, yet it can trigger replanning. How is this mechanism designed, and how does memory influence the decision-making loop in practice?

3. Section 3.5 emphasizes the importance of memory updates but does not explain how they are performed. Could the authors provide concrete details or pseudocode on the update process?

4. The paper omits important environmental setup details, such as the agent’s observation format (e.g., HTML-based or multimodal) and how actions are interfaced with the environment. Please clarify these points to ensure reproducibility.

5. The experiments are relatively limited in both benchmark coverage and baseline comparison. Are there plans to include broader evaluations or additional baselines to validate generality and robustness?

---

### Official Review · Reviewer_Lbsj · 2025-11-09

**Soundness:** 2
**Presentation:** 2
**Contribution:** 2
**Rating:** 4
**Confidence:** 3

**Summary:**

The authors propose ATLAS, a web agent consisting of LLM based modules like Planner, Actor, Critic, and Memory interacting with each other to perform a simulated look ahead tree search before selecting the final action. The memory is multilayered consisting of working memory capturing current context, cognitive map encoding state action transitions, and semantic memory capturing environment specific details, constructed through MCTS style exploration and LLM based summarization. On a simulation environment, WebArena Lite consisting of realistic web navigation tasks ATLAS demonstrates overall state of the art performance.

**Strengths:**

1. A modular agentic architecture, ATLAS, is proposed for web navigation tasks like content retrieval, form completion etc, consisting of modules for planning, memory, actor and critic.
2. Simulated look-ahead tree search using a prebuilt cognitive map through exploration, which is also used for dynamic replanning and memory update.
3. Overall state of the art performance on WebArena Lite environment.
4. Ablation studies demonstrate the effectiveness of different components.

**Weaknesses:**

1. There are no analysis of failure modes, specifcially which modules are primarily bottlenecks.

2. The overall system seems costly given that multiple modules are involved with multiple LLM calls and there is a lookahead tree search. Some cost analysis and comparison with other baselines is missing.

3. There is no improvement for multi-site tasks, raising concerns whether this approach would generalize to more complicated tasks.

4. Some key details and analysis are missing like how many action candidates are generated per step, what is the length of the simulated trajectory D, epsilon for replanning, and how sensitive is the overall method to these values?

**Questions:**

1. How is transition uncertainty U computed?



2. How does Base + CM + HL perform to justify the importance of look-ahead?

3. Some prompts like for replanning and cognitive map are missing from the appendix. Also the prompts seem to be cut out from the right margin.

4. Results tables are not referred to in the manuscript.

5. It seems some of the baselines reported in Table 1 use a different LLM than the one used in the ATLAS which is Claude 4 Sonnet? For fair comparison all methods should use the same LLM.

6. Are any incontext learning examples used for any of the modules?

---

### Note · Program_Chairs · 2026-01-17
**Submission Desk Rejected by Program Chairs**

The following references in this submission do not refer to real documents and/or have major errors in bibliographic information:

 Zhengyi Liu et al. Reinforcement learning for web information extraction. arXiv preprint arXiv:1801.07579, 2018.
Vedant et al. Sodhi. Step: Generalized planning for text-based tasks. arXiv preprint arXiv:2307.00000, 2023.